# Natural Lighting in Historic Houses during Times of Pandemic. The Case of Housing in the Mediterranean Climate

**DOI:** 10.3390/ijerph18147264

**Published:** 2021-07-07

**Authors:** Carmen Muñoz-González, Jonathan Ruiz-Jaramillo, Teresa Cuerdo-Vilches, Maria Dolores Joyanes-Díaz, Laura Montiel Vega, Victor Cano-Martos, Miguel Ángel Navas-Martín

**Affiliations:** 1Escuela Técnica Superior de Arquitectura, University of Málaga, 29071 Málaga, Spain; jonaruizjara@uma.es (J.R.-J.); lolajoyanes@uma.es (M.D.J.-D.); lmvyku0@uma.es (L.M.V.); victor.cano.martos@gmail.com (V.C.-M.); 2Instituto de Ciencias de la Construcción Eduardo Torroja, (IETcc), Consejo Superior de Investigaciones Científicas (CSIC), 28033 Madrid, Spain; 3Escuela Nacional de Sanidad, Instituto de Salud Carlos III (ISCIII), 28029 Madrid, Spain; manavas@isciii.es

**Keywords:** daylighting, residential heritage, COVID-19, lockdown, climate change, simulation, survey, circadian rhythm

## Abstract

The pandemic generated by the SARS-CoV-2 virus has led to a forced increase in the number of hours spent at home. In many cases, the lockdown situations, both in social and work terms, have meant that homes have suddenly also become workplaces. Based on all the indicators, this new relational scenario in the labor market displays a clear upward trend and is far from being a temporary situation. It is known that sunlight affects people’s circadian rhythm and that its reduction and even absence during this period of isolation has had a psychological impact on the population. This makes it necessary to reconsider the regulations applied in homes, in order to guarantee their habitability, given their recent widespread use as offices, as well as domestic spaces. In historic centers, the comprehensive renovations being carried out include improvements in energy efficiency and thermal comfort, which play a fundamental role. However, the energy consumption linked to artificial lighting and the quality of this lighting itself have remained in the background, as improvement strategies consist mainly in the replacement of incandescent or fluorescent lamps with LED lamps. Prior to the pandemic, the electric consumption of lighting systems accounted for 10–15% of the total, a figure which increased to 40–50% during the lockdown period. Aiming to improve people’s well-being while reducing energy expenditure on lighting, this article presents a quantitative approach to improving the levels of natural lighting in residential heritage buildings located in historic centers. According to data obtained from previous surveys of a sector of the population, homes built prior to 1950 were characterized by good natural lighting conditions and a very low incidence of health issues among occupants compared to contemporary homes. The objective was to quantify the circadian stimulus and lighting levels and to identify the areas or work areas in homes in order to optimize consumption related to lighting and to generate healthy and comfortable spaces. Results show that historic homes have enough naturally lit areas to perform office work during business hours. However, in the most unfavorable seasons, winter and autumn, it is necessary to use artificial lighting at the start and end of the working day.

## 1. Introduction

Regulatory initiatives for the energy efficiency of buildings sometimes exclude historic buildings given the difficulty of applying measures which do not compromise their architectural values. There are currently several lines of research at the European level on the reduction of CO_2_ emissions in these building typologies [1]. These studies aim to promote good practices in rehabilitation and conservation processes while including more sustainable improvement measures in existing buildings [2].

In many cases, the lighting of residential and office spaces represents a high percentage of energy consumption. Prior to the pandemic, the percentage of electric energy dedicated to lighting in residential buildings was between 10–15% of total consumption, a figure which reached up to 50% in office buildings [3]. Therefore, in both cases, there is great potential for energy and economic savings in the use of natural light. The labor situation brought about by the restrictions imposed by the COVID-19 pandemic has translated into a major increase in remote work and work-from-home situations, in turn increasing the energy consumption associated with the provision of lighting in homes [4]. In addition, an immediate concern arising from the continuous succession of waves of contagion has shifted attention away from other impacts associated with the pandemic. This is the case of social or economic impacts [5], which are more difficult to quantify in the short term, as research still focuses on other anthropogenic consequences such as the current grave climate emergency which is conditioning our way of life and how buildings are constructed. Unsurprisingly, it has been observed that the level of residential rehabilitation and the good condition of homes are elements which serve to protect people’s health in the face of extreme climatic events due to climate change [6].

The pandemic has revealed the importance of controlling environmental conditions in homes for people’s daily physical and emotional well-being [7]. Living spaces are not merely shelters to provide protection from the outside environment for a few hours but also spaces where human life develops. Since mobility restrictions were imposed to prevent the spread of COVID-19, companies have had to implement emergency solutions to allow their employees to work from home. Despite the progressive adaptation to normality, some experts predict that telecommuting is here to stay. In this respect, there has been a climate of widespread trust between employees and employers, fostered by a perception of high levels of productivity and performance during the lockdown period. However, this widespread perception also allows for other nuances derived from sociodemographic factors and the presence of children [5,8]. In order to promote worker safety in the face of the spread of COVID-19, many companies continue to promote teleworking as a regular work formula [5]. This modification of habits in the personal sphere due to lockdown requires dwellings to be redesigned to adapt them to these new needs. The use of natural materials, control of indoor air quality or levels of natural lighting, and access to habitable outdoor spaces are all factors that have a direct impact on people’s well-being [9]. The influence of sunlight, linked to periods of light and darkness, on the daily or circadian rhythm of the human being is well known [10]. For instance, its absence entails a lower capacity of the human body to produce vitamin D through the ultraviolet radiation of the sun. In addition, a dysregulation of the circadian rhythm can lead to lowered levels of serotonin, a molecule associated with the appearance of physical fatigue, states of sadness and, in extreme cases, depression [11]. The melatonin hormone, whose production depends on serotonin, is used as a marker for the activation of the circadian system. Thus, the circadian stimulus (CS) metric is used to quantify the impact of natural light on the melatonin level [12]. The light level, spectrum, timing and duration of exposure, and photic history should be considered. This metric is based on a model of how the retina converts light signals into neuronal signals for the circadian system [13]. CS makes it possible to evaluate the photopic illuminance provided by any light source and its effectiveness in stimulating the human circadian system, taking an exposure time of one hour as a reference. The effectiveness of the spectrally weighted irradiance at the cornea can go from threshold (CS = 0.1) to saturation (CS = 0.7). Exposure to a CS of 0.3 or higher in the eye, for at least an hour in the early part of the day, is effective in stimulating the circadian system and is associated with improved sleep, behavior, and mood. A combination of artificial and daylight sources can be used to achieve high circadian stimulation during the day and low at night.

There is a wide range of studies analyzing the potential of natural lighting to control the circadian rhythm in administrative spaces or offices [14]. This research is considered especially important given the amount of time that people spend in these spaces and the effect of alterations of the circadian rhythm on well-being, which in turn affects performance and productivity levels [15]. At the work level, one of the main effects of the COVID-19 pandemic is the increase in the number of hours of the working day that occur in the domestic environment. As people spend more time in their homes working, it has become essential to carry out an analysis of the consequences of natural lighting in these spaces. Natural light regulates human physiology and behavior, and humans spend more than 90% of their waking hours indoors. However, light in the built environment is not designed to affect circadian rhythms [16].

The COVID-19 pandemic has caused many countries to impose a curfew while the subsequent alteration in people’s daily work and social hours has also reduced their exposure to sunlight. In relation to this, a survey-based study carried out in several Chinese cities has classified the psychological impact of the pandemic as moderate-severe. Thus, 54% of those surveyed have shown symptoms of depression, anxiety, and stress [17]. This situation has also worsened while stay-at-home orders have been in place, since in many cases, contact with the outside has been exclusively through the windows, which has further contributed to a worsening of mental and emotional health conditions [18]. Studies carried out by the Institute of Construction Sciences of the Spanish Higher Council for Scientific Research (CSIC) for a project on the COVID-19 lockdown, housing, and habitability [COVID-HAB], established that a high percentage of people wanted to change natural and artificial lighting in their homes, as well as their connection with the outside world, while greater value was attached to houses with gardens or terraces [19].

The earliest first references to the incorporation of natural lighting in rooms in a standardized manner are found in the treatise “De architectura” (1st century BC) by Marcus Vitruvius Pollio [20]. This work provides instructions for solving the problem of the presence of obstacles which prevent the natural lighting of the rooms. From the section considered most appropriate for the light to penetrate, a line is drawn from the top of the wall that obstructs the passage of light to where it is needed, and if a wide space of sky can be seen when looking up from this theoretical line, the light will have no problem reaching that point. However, according to some studies [21], the invention of artificial light and its extensive use in living spaces meant that the control of natural lighting conditions indoors definitely took a back seat.

Another important consequence of lockdown among the population has been the reduction in greenhouse gas emissions. According to some studies, in Europe the greatest drop has been observed in Luxembourg with 44.6%, while the average for the rest of Europe is around 30% [22]. The decrease in the number of commutes to workplaces accounts for half the decrease in emissions in this period. Many experts argue that this could be an opportunity to make lasting changes, especially as regards to labor mobility, minimizing its impact on climate change and air quality [23]. In this regard, telework has turned out to be a beneficial option, although not without drawbacks. It also presents negative factors such as the increase in physical problems derived from a sedentary lifestyle, in addition to the alterations linked to the change in circadian rhythms mentioned previously. Just as in a conventional office, the workplace at home must comply with the requirements established by the protocols for the prevention of office occupational hazards [24].

In Spain, 28% of the real estate stock predates 1950 [25]. However, if we analyze the age of buildings in the historic centers of the cities, this percentage is much higher in the historic center of Málaga (Figure 1). The typological and constructive characteristics of a large number of buildings, built before 1950, require further study to specify the optimal environmental conditions for their use as living spaces adapted to teleworking. Adequate natural lighting conditions, combined with the reduction in emissions generated by less travel to workplaces or the decrease in energy consumption due to the use of artificial light [26], contribute to improving user health and reducing climate impact, while also significantly affecting the decarbonization of historic centers.

This article analyzes and quantifies the level of natural light in historic homes and its role in times of pandemic. The main objective is to determine the suitability of these homes for meeting the necessary requirements for hosting teleworking, minimizing the use of artificial lighting and its consequent impact on the circadian rhythm. This in turn would prevent an unnecessary increase in energy consumption and associated emissions. The specific objectives set out were: (a) to analyze Spanish homes (historic or otherwise), in terms of natural lighting and design criteria for openings, correlating them with possible productivity and circadian rhythm disorders; and (b) to analyze the specific cases of historic homes in the historic center of Málaga (Spain), to ascertain whether these types of homes meet the requirements of natural lighting for teleworking, minimizing the use of artificial light as much as possible.

With this, this study seeks to contribute to the research in two aspects: (1) through the categorization of natural lighting in historic homes and that of its light quality for tasks such as teleworking, and (2) to point out the importance of good design and dimensioning of openings in the façade.

## 2. Materials and Method

The starting point for this research is the previous analysis of the existing historic houses in the historic center of the city of Málaga, establishing the predominant building typology and the characterization of their energy behavior [27]. Subsequently, natural lighting was analyzed in order to determine its impact on people’s health, evaluating the effect of staying indoors together with having to telework from home due to the pandemic situation. The methodology used in this study makes it possible to assess whether the natural lighting of historic homes is sufficient to carry out administrative tasks and in turn minimize the use of artificial lighting. This approach to natural lighting in homes, specifically in relation to the task of working from home, was addressed from two different angles, using analysis techniques.

### 2.1. Phase 1: Assessment of the Impact of Natural Light in Homes (Survey)

This first phase is based on surveys carried out during the lockdown period to determine the incidence of lack of natural light on people. More specifically, certain qualities of the dwellings and their openings were evaluated, as well as certain disorders relating to productivity and others associated with alterations in the circadian rhythm of the people spending extended periods in them during lockdown.

For this, surveys based on an online questionnaire were conducted from 8 April 2020 to 7 June 2020, the peak period of the COVID-19 epidemic in Spain. In this country, stay-at-home orders were in place from 14 March to 21 June 2020. The questionnaire, which was limited to just nine questions, was self-completed online. This was a response to several considerations: firstly, an interest in obtaining very specific answers, ignoring sociodemographic information, housing qualities, and physiological aspects or disorders, requesting only the information most relevant to the research. Secondly, although this was not a probabilistic study, as many responses as possible were sought. Thirdly, given the rise in this type of research due to the lockdown situation and the consequent impossibility of interviewing the participants by other means, many studies emerged during this period and efforts were made to cause the least disturbance possible to interviewees.

The design of this questionnaire provides in-depth information on different aspects relating to lighting within the different types of dwellings and to the consequences of their absence during the lockdown period. Given the limited time available for the study, this synthetic questionnaire allowed us to achieve a general vision of users’ perceptions of their homes.

Thus, based on the various theoretical and practical documents, the survey drawn up concisely and efficiently covered aspects of housing, surroundings, and habitability [28,29]. In this questionnaire, objective variables of the residential environment considered included the size of the house, year of construction, and surface area of window openings, as well as other sociodemographic variables such as age, and subjective variables regarding the indoor environment, such as health consequences in the pandemic period (Table 1).

Specifically, the variables addressed and their classification by category were the following:

The link to the questionnaire was delivered to more than 1000 people throughout Spain using non-probability snowball sampling. This was disseminated through social networks, institutional websites, and instant messaging applications (such as WhatsApp^®^). The free online platform OnlineEncuesta.com (accessed on 8 April 2020) was used to collect data.

All the necessary measures were taken to guarantee respondents’ anonymity and no personal data were collected. The research team followed the recommendations of the Declaration of Helsinki at all times, both during the design of the survey and the treatment of the information. Therefore, the participants were informed beforehand about those responsible for the research project and the objective of the study, and how they could freely and anonymously access the online form, thereby providing their implicit consent.

### 2.2. Phase 2: Analysis and Quantification of Natural Light in Historic Homes through the Application of Computer Programs

The second phase of this study was based on the light characterization of different types of historic housing using virtual models. Simulation software such as DIAlux evo and Designbuilder, which works with the Daysim 4.0 calculation engine, was used to qualify and quantify the natural lighting of these spaces. This tool combines a daylight coefficient approach with the Perez all-weather sky model [30] to predict the amount of daylight in and around buildings, based on direct normal and diffuse horizontal irradiances taken from a climate file. Several study scenarios were designed, taking into account the different seasons of the year, spring, summer, autumn, and winter, as seasonal changes influence the availability of natural light. For the usage profile, working hours, Monday to Friday, from 8:00 a.m. to 18:00, were taken into account. The meteorological data for the location were obtained from the Energyplus 8.9 calculation engine, developed by the US Department of Energy [31] and the Spanish National Institute of meteorology (AEMET).

The metrics used for this study for the effect of natural lighting and the analysis of energy consumption were the circadian stimulus (*CS*), daylight autonomy (DA), and annual solar exposure (ASE).

CS is mainly related to spectral power distribution (SPD) and illuminance perceived by users and circadian light (CLA). This can be calculated through the following Equation (1) [14]:(1)$$CS=0.7\left(1-\frac{1}{1+{\left(\frac{\mathrm{CLa}}{355.7}\right)}^{1.1026}}\right)$$

In the case of housing, the *CS* of the spaces was analyzed through the average lighting levels during working hours to establish whether these spaces provide the amounts prescribed to be healthy in the event of having to work remotely [32].

Another metric used to determine energy savings is daylight autonomy (DA) [33], defined as the percentage of busy hours of the year (working hours), when the minimum illuminance threshold of 300 lux is reached in a work plane [34]. The higher the DA value, the lower the energy consumption associated with switching on artificial lighting.

Finally, the Annual Solar Exposure (ASE) was analyzed, based on the Lighting Measurement 83 test [35] and the calculation of the IESNA (Illuminating Engineering Society of North America) was applied [36]. Annual solar exposure defines how much of the space receives too much direct solar radiation, which can cause glare discomfort in teleworking tasks where the equipment used includes screens and computers. The unit of measurement is the percentage of hours that the illuminance exceeds the upper limit of 1000 lux. The goal of ASE 1000/250 is for there to be 250 h or less in which that limit is exceeded. To reduce the potential for glare and heat stress, designers should aim for low values. Visual discomfort can occur with values greater than 10% (1000 lux are exceeded for 250 h).

The necessary validation, derived from the use of calculation tools, was carried out, taking as reference similar studies developed with the Daysim 4.0 calculation engine (Massachusetts Institute of Technology, Cambridge, MA, USA), from which dynamic metrics can be calculated with precision for spaces with similar sizes and boundary conditions [37].

### 2.3. Historic Housing in the City of Málaga

In this research, two historic houses in the historic center of the city of Málaga (LAT. 36.72° N LONG. −4.42° W) were analyzed. Different predominant typologies in the study area (single-family and multi-family) were selected in order to carry out a comparison of the lighting conditions (Figure 2 and Table 2).

The four-story multi-family dwelling dates from 1920, has north- and west-facing facades, with a rectangular floor plan (7.20 × 10.20 m^2^) and a total surface area per floor of 74 m^2^. The single-family dwelling dates from 1936 and has a north and south facade. Its rectangular plan (8.95 × 16.73 m^2^) is aligned with the road separated from the front garden by a dividing wall. It has two volumes separated by a cornice, with the openings arranged in three symmetrical rows. Its total surface area per floor is 123 m^2^. None of the houses analyzed currently have indoor courtyards as all the original ones were eliminated in different rehabilitations.

The multi-family dwelling has a window-to-facade ratio of 34%, while the single-family dwelling has a window-to-facade ratio of 22%. Table 3 below shows the surface area of walls and openings in relation to their orientation.

## 3. Results

### 3.1. Phase 1: Evaluation of the Impact of Natural Light in Homes through Surveys

A total of 1159 people accessed the questionnaire. Of these, 838 questionnaires were submitted and their responses considered valid.

The survey was carried out among a population aged 18–70. In addition, 40% of respondents were between 20 and 40 years old, 40% between 41 and 65 years old, 5% were over 65 years old, and 7% were aged 18–20.

These responses provided data on the Spanish population where 85% lived in latitudes between 35–40°, 14% in latitudes between 40–44°, and 1% in latitudes above 44°.

Of the responses obtained, 9.7% related to historic homes (predating 1950), while 19% corresponded to homes built between 1950 and 1979, 48.3% to homes built between 1980 and 2006, and 23% to homes built between 2006 and 2020.

Of the users surveyed, 55% lived in multi-family housing and 41% in a single-family home, of which only 16% had garden spaces. Further, 4% of those surveyed lived in other types of housing, such as apartments. The surface area of the spaces analyzed also varied greatly. Specifically, 60% of them had a surface greater than 90 m^2^, 30.3% between 70–90 m^2^, 8.4% between 46–70 m^2^, and only 1.3% between 45–20 m^2^ (Figure 3).

As regards to the uncovered spaces of the house in which users could have direct contact with natural light, 78.4% of the houses had a patio or balconies, while the remaining 21.6% did not. Likewise, in relation to the type of opening, 68.3% of the dwellings had balconies.

Figure 4 provides a rundown of the health disorders detected during the time spent at home during lockdown. Of the people surveyed, only 22% suffered no disturbance. Among the disorders analyzed, the most prevalent were headaches, exhaustion, and lack of concentration. Likewise, a classification of health disorders by building type showed that 57–64% of those living in multi-family dwellings reported headaches, eye pain, and lack of concentration. These disorders decreased by up to 15–25% among those living in single-family homes and by 12–15% in single-family homes with a garden.

Of the people surveyed, 58% were under 20 years of age and had some type of ailment. Those most affected were in the 20–40 age range, with 72% reporting some type of discomfort. In the case of the age groups of 41–65 and over 65, 48% and 61% reported disorders, respectively.

The correlation between the year of construction of the house and the annoyances or disorders reported by respondents was analyzed. Incidence was 8% in the homes built prior to 1950, 37% in homes built in the 1950–2006 period, and 25% in homes built after 2006.

#### Bivariate Relationships between Variables

To establish the relationships between the study variables, Pearson’s Chi-square test was applied, establishing a statistically significant relationship if the p-value was less than or equal to 0.05.

The age of the dwelling had a significant relation to the presence of vertical windows (*p* = 0.000), either exclusively or combined with horizontal ones. The historic houses (pre-1950) and the most modern ones (after 2006) were those that presented this type of configuration of openings. As regards to disorders, the age of the home was shown to be linked to headaches (*p* = 0.000) while historic homes did not generate this type of ailment.

The absence of vertical windows, either exclusively or alternating with horizontal ones, was statistically related to reports of this type of disorder in isolation or in combination with others (*p* = 0.003), specifically lack of concentration (*p* = 0.005).

The surface of the house showed a significant relationship with certain disorders such as headaches (*p* = 0.000), lack of concentration (*p* = 0.001), and exhaustion (*p* = 0.001).

Similarly, not having a patio or terrace was statistically related to suffering from headaches (*p* = 0.001), lack of concentration (*p* = 0.001), or exhaustion (*p* = 0.005). Eye disorders did not show a significant relationship with whether or not these spaces were open to the outside.

Regarding the possibility of experiencing more than one of the disorders mentioned above, the variables that showed a statistically significant relationship were: living in houses built after 1950 (*p* = 0.000), multi-family or single-family terraced (*p* = 0.007), not having a patio or terrace (*p* = 0.007), and living in less than 90 m^2^ (*p* = 0.000).

The presence of balconies in the dwellings was not statistically related to any of the disorders exposed or to their combination.

### 3.2. Phase 2: Characterization of Natural Lighting through Computer Simulations

The results obtained from the computer simulation of the average illuminance in summer and winter at different times of the day (corresponding to working hours, from 8:00 a.m. to 18:00) and for different floor heights are shown below (Figure 5).

The multi-family dwelling displays high lighting levels, as there is a larger surface of openings, almost 12% more than in the single-family dwelling. The average lighting level in multi-family dwellings is around 1200 lux in summer and 443 in winter, with the highest values being found at 14:00–18:00. In the case of single-family homes, the average levels are between 225 lux in summer and 95 lux in winter, while the maximum values are observed at 12:00–16:00 in the afternoon (Figure 6).

The circadian stimulus, a transformation of circadian light on a relative scale, was analyzed based on lighting levels. The recommended values are 0.3–0.4 during the morning, with a high content of short wavelengths (as is the case in daylight).

According to the results obtained in summer and winter, it is established that these limits are higher during all working hours in the case of multi-family housing. This is not the case in winter hours where the adequate levels occur at 10:00–16:00. In the case of single-family homes, the levels would be suitable for summer from 10:00–18:00 and in winter from 12:00–15:00.

According to the data obtained from DA with a minimum illuminance threshold of 300 lux during working hours, the average value of DA on all floors is 70% of the time in the case of multi-family dwellings. As Figure 7 shows, the values are around 80–95% of the time in the areas near the window openings and in the central area of the dwelling. For single-family homes, the average DA value is lower, 28% on the ground floor and 40% on the first floor. However, in the areas near the window openings, the percentage is higher, 80–95%. When DA presents values of 70% or higher, users tend to prefer natural lighting to artificial lighting, as they consider it adequate to work comfortably.

The data obtained from ASE, an indicator which determines that there is no risk of glare or overheating due to natural lighting, show how the average value of hours exceeding 1000 lux is around 100–150 h in the case of the multi-family dwelling, and 42–60 h in the single-family dwelling (Figure 8). These parameters determine the areas in homes with DA levels adequate for optimizing the use of natural light and to be able to telework without any problem of discomfort due to glare or solar charge, since the percentage of the ASE parameter is minimal.

Considering the three parameters analyzed, it is possible to establish the most suitable zones or areas for the location of the work zones in the homes to have an impact on the circadian rhythm, saving energy by the use of natural light under conditions of light and thermal comfort. The zones and percentages of the same with DA levels lower than 70% are shown, with ASE higher than 10% and areas suitable for telework, which require DA > 70%, ASE < 10%, and CS > 0.3.

As Figure 9 shows, multi-family housing has a larger area where it is possible to telework, between 58–68% (Table 4). In single-family homes, this drops to 23–35%. In both types of dwellings, the adequate surface for work increases along with their height.

## 4. Discussion

This study analyzed and quantified the natural lighting of homes, especially historic homes, using two different levels of approximation. The fundamental objective was to determine the suitability of historic homes for teleworking in terms of natural lighting and its impact on circadian rhythms, especially in times of a pandemic. As a specific objective, it was proposed to determine the most appropriate design and type of opening for these requirements, as well as the area of the plans that can usually be occupied and where natural lighting is adequate for work tasks. At present, we only find this type of study for health [38], educational, or office [39] spaces and linked to standard or horizontal window openings. According to the results obtained, the absence of horizontal windows and the availability of patios or terraces result in the highest incidence for potentially experiencing some type of disorder due to the lack of lighting. Likewise, access to a large living space (greater than 90 m^2^) or a single-family home are some of the domestic characteristics with a positive impact on avoiding these disorders.

According to the surveys carried out and the data obtained from the simulation, historic homes display good lighting qualities with a lower incidence of productivity-related disorders and other annoyances. In fact, no headaches or other disorders were reported in homes built prior to 1950.

Applying this methodology in historic homes can quantify energy savings or, in future renovations, provide design guidelines for the use of daylight to improve the health and well-being of the occupants, even incorporating artificial lighting. This is an instrument that provides a means of identifying the areas of a floor plan which present the greatest potential for natural lighting with a circadian effect. Figure 9 shows the areas where the occupants would depend on artificial lighting systems. This can also serve as a manual for the interior design of a project, zoning, and the operation of natural and artificial light.

The application of this methodology in this document was limited to two case studies, representative of the typology to be found in the historic center of Málaga. However, further research, using a broad set of variables (e.g., orientation, climate, latitude, material properties, etc.) is needed to explore and improve understanding of their applicability.

The results do not take into account socioeconomic status or family size. The primary variables that were used to normalize study participants were home type and size. It is a factor that will be take into account in future research. Other studies carried out determined the importance of daylight in the time of a pandemic. The people looked for the most illuminated areas to telework for in their homes [40,41,42].

This research has not considered stressors related to share building multi-family (e.g., noise associated with shared walls), because that contemporary home, according to regulations, has better acoustic conditioning, and the results showed that these home present a higher percentage of discomfort, disorders, etc.

There is currently a debate on what the appropriate level of light stimulation should be within buildings to maintain a healthy circadian entrainment, and how wavelength factors should be considered. However, there is a general consensus that naturally illuminated spaces help circadian readjustment in the morning, a fundamental factor for the health and well-being of the building’s occupants. The electric lighting industry actively promotes color-adjusted artificial lighting as an effective substitute for sunlight and sky. However, there is an interest in natural lighting to achieve green building construction or rehabilitation certifications, health-and-wellness and energy-focused certifications. Hence, the need for this design tool to help technicians and designers to use daylight as a source to meet human circadian lighting needs in buildings.

The 838 validated surveys have allowed this study to provide a set of rational results on the incidence of natural light in different types of historic housing. This result confirms that the survey can be a practical way to obtain data from the residents who live in this building typology and show their preferences. This study can provide guidelines on the causal relationship between ailments and the lack of natural light, considering physical aspects and spatial conditions.

This document is expected to carefully consider the subject of workspace in historic homes and health issues related to natural lighting, making this information known to the technicians. Spaces of architecture becomes an important component in this discussion given that all parameters of daylight (intensity, time, and spectrum) are determined through the form of the surrounding structure. Light intensity is conditioned by the size and shape of the openings.

## 5. Conclusions

After the period of lockdown due to COVID-19, homes have become another space from which to telework. It has therefore become necessary to reconsider their indoor environmental conditions and thus, the regulations applied to guarantee their habitability and comfort in multiple functionality situations. This situation has led to a revolution in the way of working, normalizing teleworking. This situation can have beneficial effects on climate change due to the reduction in CO_2_ emissions linked to lower levels of work commutes. This normalization of telework means that a high percentage of the population spends more hours at home, making it necessary to analyze their environmental conditioning.

In addition, the comprehensive rehabilitation of historic buildings of heritage homes begins to consider as relevant criteria the improvement of energy efficiency and thermal comfort, aspects that also affect the decarbonization of historic city centers. However, the lockdown situation derived from the pandemic has highlighted the inescapable need to consider other parameters such as natural lighting in design. This, in addition to being a benefit for people’s health, involves a reduction in energy consumption related to the lesser use of artificial lighting. In addition, sunlight affects the circadian rhythm of people and during the period of isolation its absence has had a psychological impact on the population.

In this research, the natural lighting of living spaces has been analyzed in order to establish strategies for the improvement and adaptation of these rooms for a new function, teleworking. For this, lighting levels, their impact on the circadian rhythm, energy consumption/saving, and habitability have been analyzed.

From the current analysis of the buildings, most of the houses have spaces or elements that allow contact with natural light. However, during the lockdown period, most people reported some type of ailment or health problem. Of the disorders analyzed, those with the highest incidence were headaches, exhaustion, and lack of concentration.

It is interesting to relate the building type and year of construction of the dwellings with the health problems presented by the users. People who lived in multi-family homes showed more health problems than those who lived in single-family homes. In addition, residents of single-family homes with garden spaces or with access to natural light hardly presented any disorders.

People who lived in homes built prior to 1950, slightly manifested some kind of discomfort, however, health problems increased in homes built more recently. This factor shows the importance of quantifying natural light during working hours, in different types of historic houses built prior to this year.

The lighting levels of these houses, as expected, were higher in those that had a greater surface of window openings. In both cases during the summer season, the levels were adequate for the circadian stimulus. However, in winter both early and late in the morning, it was not adequate, so it would be necessary to use artificial light to implement this deficiency.

This typology of historic homes shows spaces or enclosures which, for at least 80–95% of working hours, do not need the use of artificial lighting to be able to carry out work or training tasks, so only between 5–20% of the time would require the energy consumption associated with artificial lighting. These homes have areas suitable for locating work areas, where the lighting levels are adequate to maintain the circadian rhythm, perform telework or training tasks, and without problems of glare or overheating for the user. This supposes an energy savings related to the use of natural light in the illumination of these spaces.

This study can contribute to the improvement of climate change by saving energy from the use of artificial light, and be added as a strategy to decarbonize historic centers. The energy savings due to daylight not only offset energy consumption associated with heating and cooling, however, design strategies are established for the optimization of energy resources and health in homes, for their relevant consideration by specialists. High-density historic centers contain historic homes that have been repurposed as apartments or offices, and the presence of these structures maintains the traditional character of the area. Similarly, restricting new construction is an important step towards reducing energy use. For these reasons, it has become important to consider whether existing structures can be adapted to become shared residences and workplaces. For future studies, it would be interesting to analyze the natural lighting of contemporary homes to establish a comparison with historic homes.

## Figures and Tables

**Figure 1 ijerph-18-07264-f001:**
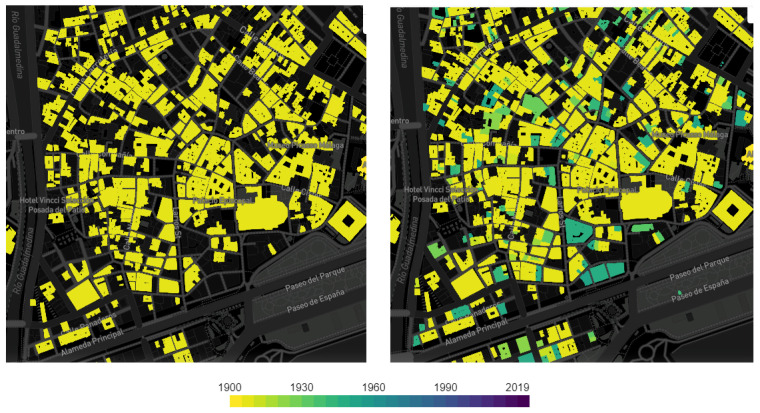
Housing age map in the historic center of Málaga [27]. © Colaboradores de OpenStreetMap.

**Figure 2 ijerph-18-07264-f002:**
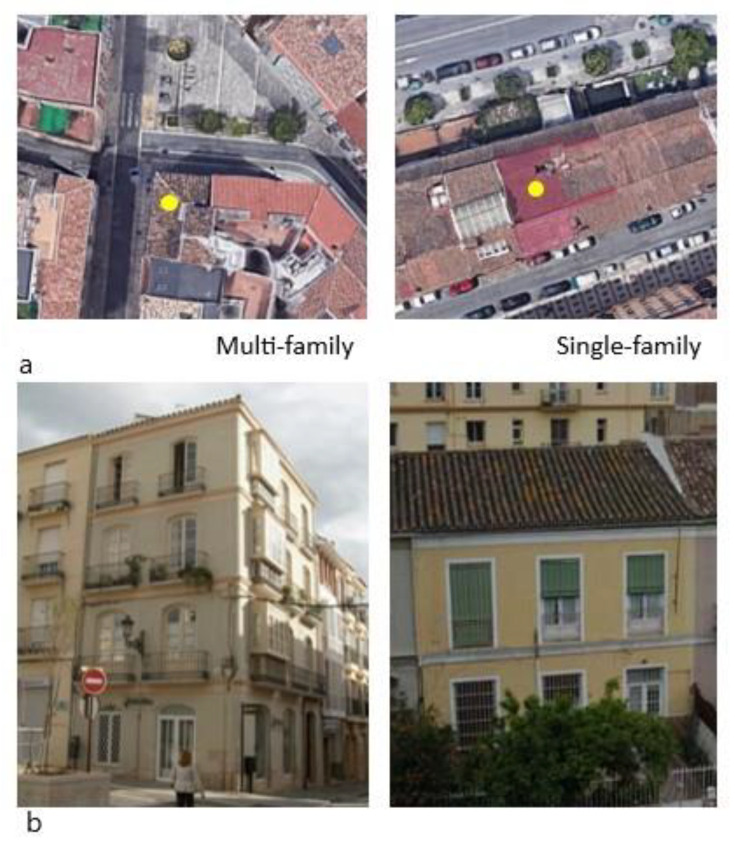
(**a**) Aerial photographs (**b**) facades photographs [27].

**Figure 3 ijerph-18-07264-f003:**
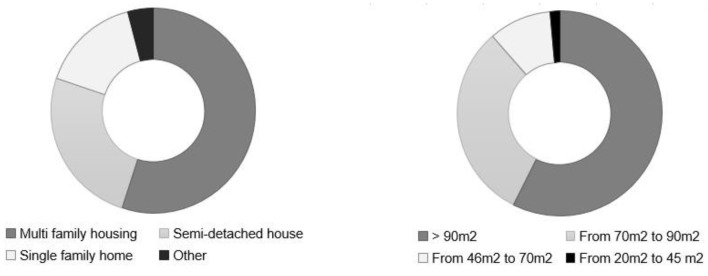
Results of dwelling typology and surface areas.

**Figure 4 ijerph-18-07264-f004:**
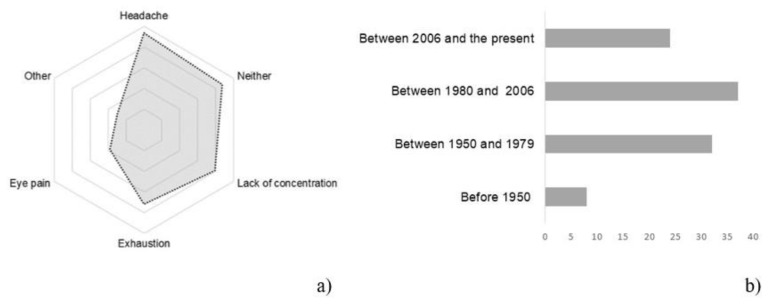
(**a**) Results of disorders during the stay-at-home period; (**b**) Cumulative percentage of disorders related to construction years of housing.

**Figure 5 ijerph-18-07264-f005:**
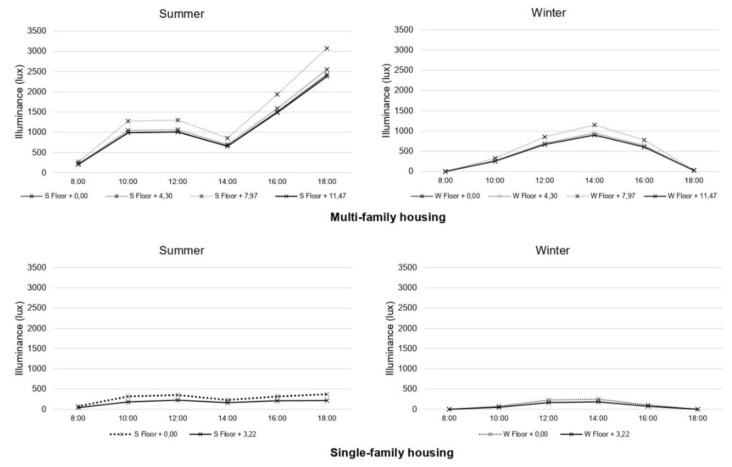
Average lighting level in homes during different seasons.

**Figure 6 ijerph-18-07264-f006:**
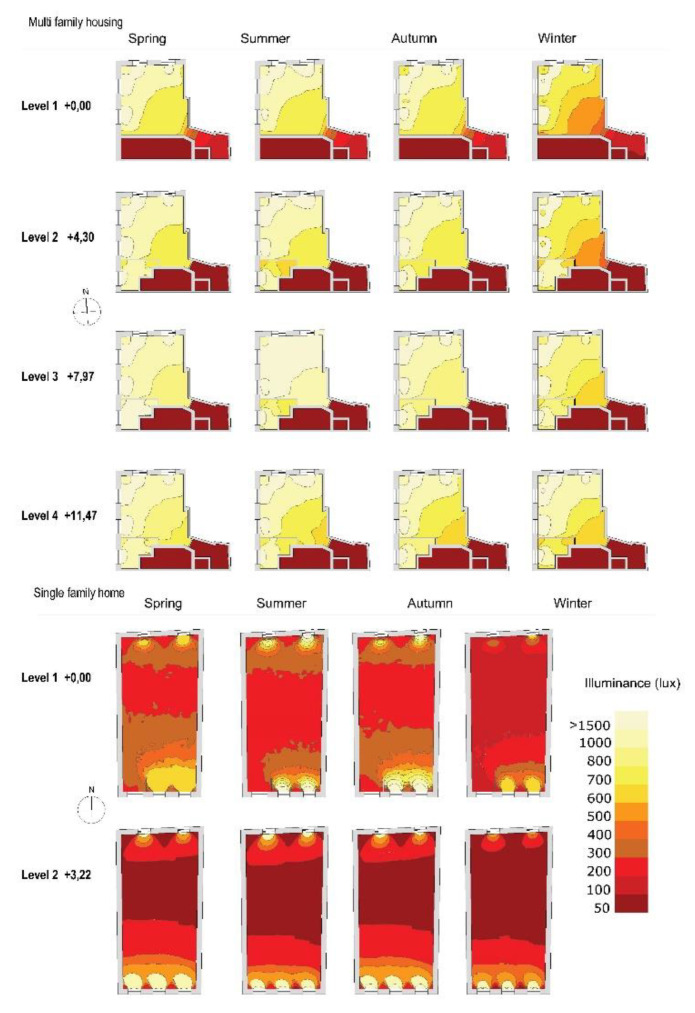
Lighting level in homes during different seasons at a height of the workplane. (**top**) Multi-family housing. (**bottom**) Single family housing. Calculation time 12:00 p.m.

**Figure 7 ijerph-18-07264-f007:**
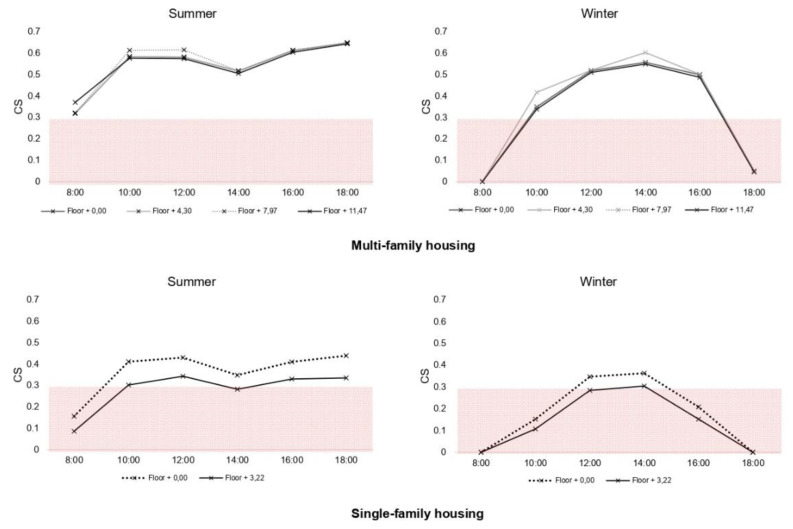
Circadian stimulus by residential typology and floor.

**Figure 8 ijerph-18-07264-f008:**
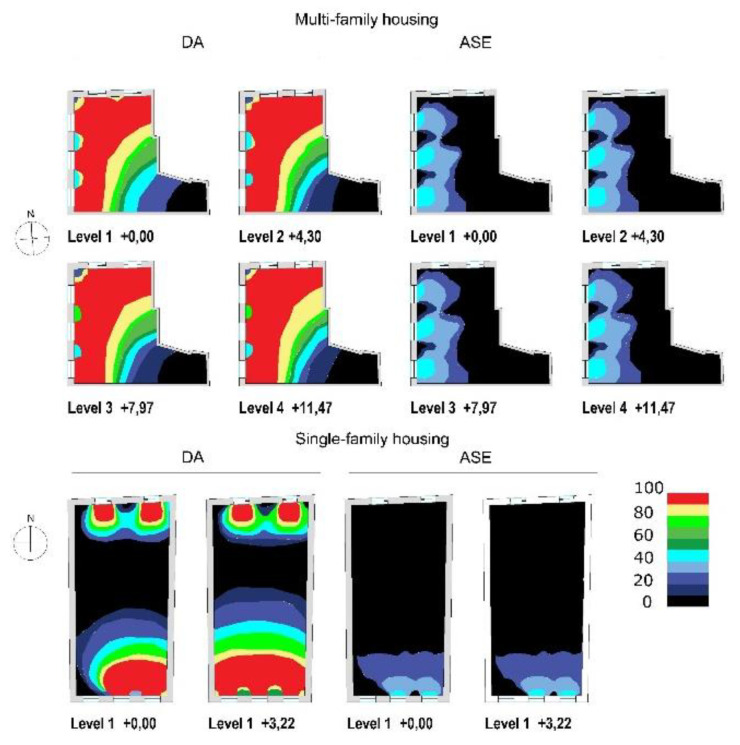
Multi-family and single-family housing results, DA (%) and ASE (%) parameters.

**Figure 9 ijerph-18-07264-f009:**
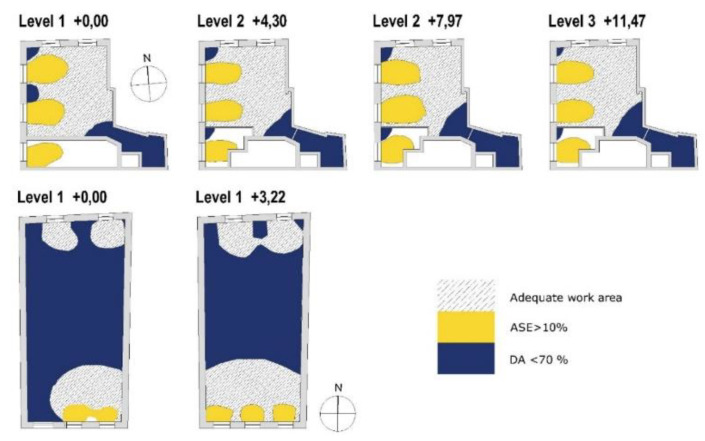
Location of suitable areas for teleworking in multi-family and single-family housing. Space conditions DA > 70%, ASE < 10%, and CS > 0.3. DA (Daylight Autonomy), ASE (Annual solar exposure), and CS (Circadian stimulus).

**Table 1 ijerph-18-07264-t001:** Categories and variables requested in the survey for evaluation.

Sociodemographic characteristics	1. Respondent’s age 2.Autonomous Community where they live.
General characteristics of the house	3.Type of home 4. Home surface area 5. year of home construction.
Openings and spaces in contact with the outside	6. Existence of patio or terrace in the house 7. Presence of balconies 8. Availability of vertical windows in the house.
Disorders suffered in lockdown	9. a) Eye disorders b) Headaches c) Exhaustion d) Lack of concentration e) Others f) None.

**Table 2 ijerph-18-07264-t002:** Characteristics of the dwellings analyzed.

Dwelling	Material Outer Walls	Material Interior Walls	Wall Thickness (m)	Reflection Coefficient (%)
Multi-family	brick wall	brick wall	0.55–0.60	Wall = 70%
Single-family	brick wall	brick wall	0.50–0.55	Floor = 40%
				Ceiling > 80

**Table 3 ijerph-18-07264-t003:** Opaque wall surface and window openings (m^2^).

MULTI-FAMILY	N	S	E	W
Wall	150	-	-	228
Window	43.75	-	-	84.3
**SINGLE-FAMILY**	N	S	E	O
Wall	55.6	59	-	-
Window	15	10	-	-

**Table 4 ijerph-18-07264-t004:** Percentage of surface where DA is less than 70% of working time, ADE is higher than 10% of working time, and percentage of surface suitable for working (DA 70%, ASE < 10%, and CS > 0.3).

Multi Family Housing	DA 300 < 70%	ADE < 10%	Surf. Suitable for Working
Floor + 0.00	21%	21%	58%
Floor + 4.30	9%	20%	71%
Floor + 7.07	11%	28%	61%
Floor + 11.47	11%	21%	68%
**Single Family Housing**	**DA 300 < 70%**	**ADE < 10%**	**Surf. Suitable for Working**
Floor + 0.00	73%	4%	23%
Floor + 3.22	60%	6%	35%

## Data Availability

The data are not publicly available due to ethical reasons.

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
