# Peer review of "Natural Lighting in Historic Houses during Times of Pandemic. The Case of Housing in the Mediterranean Climate"

_ijerph, 2021, doi:10.3390/ijerph18147264_

Round 1
Reviewer 1 Report
The authors present an interesting study on the levels of natural lighting in residential homes, based on home type, and the influence on circadian rhythm. The objectives are clear, to quantify circadian stimulus based on natural lighting levels and identify areas within the home with natural lighting in order to optimize consumption. The manuscript would benefit from a more detailed explanation of the software models used, a list of survey questions, higher resolution figures, additional proof-reading for reading clarity, and a more thorough discussion of the results based on the comments below.
Line 7: Author affiliations are missing and should be included
Lines 5-6: (formatting change) Laura Montiel Vega is at larger text size than the other names.
Line 14: insert comma after “…use as office,”
Line 16: insert comma after “…and thermal comfort,”
Line 20: insert space between sentences “…lockdown period. Aiming to…”
Line 44: “after the pandemic” should be specified after the following statement “…up to 50% in office buildings.”
Line 75: change “rays of light” to radiation
Line 77: Change “element” to molecule
Line 85: a brief explanation of the range of CS values would be helpful here
Page 3: Is there any effect on CS based on the color/temperature of light (e.g. warm vs cool colors) or output (lumens)?
Line 135: (Figure 1) the maps appear to be showing the same location, but with different ages plotted. It appears that different scales were used to create each map. The authors are encouraged to review the figure and make sure the appropriate map and corresponding scale are shown.
Line 159-162: sentence needs rephrasing for reader clarity
Line 174-176: sentence is confusing and seem unnecessary.
Page 6: the software/models used are unfamiliar to the reviewer. It is left to the editor to decide if specific parameters should be included for reference in this section.
Line 207: (Table 1) the contents of the table are bit confusing. The authors are encouraged to restructure it in an easier to understand format. Alternatively, the authors might list the 9 questions on questionnaire.
Line 237: (eq 1) The constant values (0.7, 355.7, 1.1026) should be defined in the figure caption if possible. “a” should be capitalized in the “CLa” in the equation.
Line 272: (Figure 2) the figure caption should describe each panel and the images should be labeled as “a, b, c, d”. Orientation is not shown on the figure as described in the caption. None of the images represent floor plans as described in the caption. “Aerial photographs” would be a better description.
Line 277: (formatting change) spacing should be adjusted so that Table 3 is on a single page and not split between pages 7-8.
Line 295: (Figure 3) The figure appears to be cropped and the legend is cut off on the right margin. It is difficult to distinguish between the shades of gray used in the pie charge. The authors are encouraged to remake this figure in color.
Line 315: (Figure 4) Caption needs clarification for (b). Consider adding in “cumulative” before “Percentage of disorders…”
Line 317: “Bivariate relationships between variables” Is this a section heading or an incomplete sentence that was missed? If it is a section heading, please format appropriately (i.e. 3.1.1 added before italicized title)
Line 345: (Figure 5) figure is poor quality/low resolution and should be updated. The font sizes for the titles and legends are different between each panel and should be uniform.
Line 353: (Figure 6, formatting change) the title is on a separate page from the figure. The figures are offset form one another and should be aligned for visual appeal. The lower panel showing “single-family home” appears to have text cut off at the top. The figure caption should briefly describe each figure, and the figures should be adequately explained in the text.
Line 362: (Figure 7) figure is poor quality/low resolution and should be updated. It is difficult to distinguish between the lines on multi-family housing graphs.
Line 379: (Figure 8) figure is poor quality/low resolution and should be updated. The scale values are difficult to read. Remove extra zeros on the scales for the panels on the right side.
Line 389: (Figure 9) ASE, DA, and CS should be defined in the caption
Line 391: (Table 4) ASE, DA, and CS should be defined in the caption
Line 433: “almost a thousand validated” should be changed to the actual number of surveys.
Line 439-444: Consider rephrasing this paragraph for clarity.
Line 467: “savings” were not previously discussed in the text
Figures: (2, 3, 5, 6, 8) should be referenced within the text
References: (22, 23, 28, 29) don’t have author listed and should be checked to confirm
References: (3, 4, 13, 36) don’t have page numbers listed
Reference 5: change “Join” to Joint
General comments:
- The discussion is brief and would benefit from a more thorough discussion of the implications of the results, as well as a comparison to other factors (employment status, socioeconomic status, etc.) that may cause similar factors results.
- Do the results take into account socioeconomic status, employment status, or family size? What criteria were used to normalize study participants so that the primary variable was home type?
- There is a brief mention of teleworking with children at home. Have the authors considered scenarios where two (or more) working adults are home, as well as children. The stress factors associated with low light metrics (e.g. headache, loss of focus, etc.) would presumably be impacted by sharing work spaces. Likewise, it would be impractical to share the spaces deemed ideal for work based on natural light.
- Have the authors considered stressors related to shared buildings “multi-family” (e.g. noise associated with shared walls, or concerns over contagion) that may have impacted the results?
- Regarding implications for climate change: Have the authors considered other energy consumption factors associated with teleworking (e.g. heating and cooling during the day) that may not be set/used at the same level when the home is unoccupied? Would this offset any energy savings associated with natural light?
Author Response
First, we would like to say thank you to the reviewers for the useful comments to improve the paper. We have addressed all the comments as explained below.
Reviewer 1:
The authors present an interesting study on the levels of natural lighting in residential homes, based on home type, and the influence on circadian rhythm. The objectives are clear, to quantify circadian stimulus based on natural lighting levels and identify areas within the home with natural lighting in order to optimize consumption. The manuscript would benefit from a more detailed explanation of the software models used, a list of survey questions, higher resolution figures, additional proof-reading for reading clarity, and a more thorough discussion of the results based on the comments below.
Line 7: Author affiliations are missing and should be included
The Author affiliations have been included
Lines 5-6: (formatting change) Laura Montiel Vega is at larger text size than the other names.
It has been modified
Line 14: insert comma after “…use as office and ”Line 16: insert comma after “…and thermal comfort,”
Done
Line 20: insert space between sentences “…lockdown period. Aiming to…”
It has been included
Line 44: “after the pandemic” should be specified after the following statement “…up to 50% in office buildings.”
Currently, there are no published studies on consumption and percentages. The data provided is prior to the pandemic.
Line 75: change “rays of light” to radiation
It has been changed
Line 77: Change “element” to molecule
Done
Line 85: a brief explanation of the range of CS values would be helpful here
It has been described in more detail.
Page 3: Is there any effect on CS based on the color/temperature of light (e.g. warm vs cool colors) or output (lumens)?
In paragraph 86 these data have been included.: “The light level, spectrum (color), timing and duration of exposure, and photic history should be considered “.
Line 135: (Figure 1) the maps appear to be showing the same location, but with different ages plotted. It appears that different scales were used to create each map. The authors are encouraged to review the figure and make sure the appropriate map and corresponding scale are shown.
It has been changed
Line 159-162: sentence needs rephrasing for reader clarity
The sentence has been revised
Line 174-176: sentence is confusing and seem unnecessary.
This sentence has been removed
Page 6: the software/models used are unfamiliar to the reviewer. It is left to the editor to decide if specific parameters should be included for reference in this section.
The authors consider that this data and technical specifications should appear, as it is an essential basis of the article. The means used for them and their characteristics must be specified.
Line 207: (Table 1) the contents of the table are bit confusing. The authors are encouraged to restructure it in an easier to understand format. Alternatively, the authors might list the 9 questions on questionnaire.
The table has been modified for a better understanding
Line 237: (eq 1) The constant values (0.7, 355.7, 1.1026) should be defined in the figure caption if possible. “a” should be capitalized in the “CLa” in the equation.
This has been changed. About constant values, This has been changed. About constant values, the Circadian Stimulus (CS) depends on scientific model of human circadian photo-transduction created by Rea, whose reference has been attached in the paper
Line 272: (Figure 2) the figure caption should describe each panel and the images should be labeled as “a, b, c, d”. Orientation is not shown on the figure as described in the caption. None of the images represent floor plans as described in the caption. “Aerial photographs” would be a better description.
The texts have been modified
Line 277: (formatting change) spacing should be adjusted so that Table 3 is on a single page and not split between pages 7-8.
The table has been scrolled
Line 295: (Figure 3) The figure appears to be cropped and the legend is cut off on the right margin. It is difficult to distinguish between the shades of gray used in the pie charge. The authors are encouraged to remake this figure in color.
The graphic and its colors have been modified
Line 315: (Figure 4) Caption needs clarification for (b). Consider adding in “cumulative” before “Percentage of disorders…”
It change has been considered
Line 317: “Bivariate relationships between variables” Is this a section heading or an incomplete sentence that was missed? If it is a section heading, please format appropriately (i.e. 3.1.1 added before italicized title)
3.1 has been added
Line 345: (Figure 5) figure is poor quality/low resolution and should be updated. The font sizes for the titles and legends are different between each panel and should be uniform.
The resolution of figure 5 has been improved
Line 353: (Figure 6, formatting change) the title is on a separate page from the figure. The figures are offset form one another and should be aligned for visual appeal. The lower panel showing “single-family home” appears to have text cut off at the top. The figure caption should briefly describe each figure, and the figures should be adequately explained in the text.
The figure has been improved and described.
Line 362: (Figure 7) figure is poor quality/low resolution and should be updated. It is difficult to distinguish between the lines on multi-family housing graphs.
The Figure 7 has been improved
Line 379: (Figure 8) figure is poor quality/low resolution and should be updated. The scale values are difficult to read. Remove extra zeros on the scales for the panels on the right side.
This figure has been changed
Line 389: (Figure 9) ASE, DA, and CS should be defined in the caption
Line 391: (Table 4) ASE, DA, and CS should be defined in the caption
It is understood that these terms have already been defined in the text and the reader is familiar with them (línea 240-260). Despite that it has been described again.
Line 433: “almost a thousand validated” should be changed to the actual number of surveys.
It has been changed
Line 439-444: Consider rephrasing this paragraph for clarity.
This paragraph has been considered
Line 467: “savings” were not previously discussed in the text
In this case only consumption is specified. This has been modified.
Figures: (2, 3, 5, 6, 8) should be referenced within the text
The reference of figures have been included.
References: (22, 23, 28, 29) don’t have author listed and should be checked to confirm
This has been changed, but some authors are institutions.
References: (3, 4, 13, 36) don’t have page numbers listed
These have been included.
Reference 5: change “Join” to Joint
This has been changed
General comments:
- The discussion is brief and would benefit from a more thorough discussion of the implications of the results, as well as a comparison to other factors (employment status, socioeconomic status, etc.) that may cause similar factors results.
- Do the results take into account socioeconomic status, employment status, or family size? What criteria were used to normalize study participants so that the primary variable was home type?
- There is a brief mention of teleworking with children at home. Have the authors considered scenarios where two (or more) working adults are home, as well as children. The stress factors associated with low light metrics (e.g. headache, loss of focus, etc.) would presumably be impacted by sharing work spaces. Likewise, it would be impractical to share the spaces deemed ideal for work based on natural light.
- Have the authors considered stressors related to shared buildings “multi-family” (e.g. noise associated with shared walls, or concerns over contagion) that may have impacted the results?
- Regarding implications for climate change: Have the authors considered other energy consumption factors associated with teleworking (e.g. heating and cooling during the day) that may not be set/used at the same level when the home is unoccupied? Would this offset any energy savings associated with natural light?
All the factors considerated by the reviewer are interesting for future studies.
However, this research focuses on the study of the incidence of natural lighting on people.
The development of the surveys was oriented to characterize the daylight of the different dwellings according to the year of construction and its incidence on people. The survey described the factors that were analyzed in this research. Likewise, the people surveyed could note other disorders suffered in lockdown
The age range of the respondents was characterized and it is interesting to observe that in all ages, young people and adults, there is a high incidence of ailments. Likewise, contemporary homes, according to regulations, have better acoustic conditioning, and the results show that these homes present a higher percentage of discomfort, disorders, etc.
In relation to the energy consumption of cooling and heating, a more extensive study is being contemplated. Energy savings due to natural lighting cannot compensate for the energy consumption associated with thermal conditioning, but in this case it is intended to enhance healthy spaces with the use of natural sources.
Reviewer 2 Report
The research is very interesting and is well founded and developed with a relevant methodology, which is clearly described. It is suggested to enhance the images, for example, in figure 1, which is shown in each image (right and left). Not all images are cited in the document.It is suggested to add explanatory notes on the figures to improve their understanding. Add an image of the city under study, where it is located in Spain and within it, which is the area studied.
It is not clear if all the people who answered the questionnaire live in the historic center, or are distributed throughout the city.
Author Response
First, we would like to say thank you to the reviewers for the useful comments to improve the paper. We have addressed all the comments as explained below.
Reviewer 2:
The research is very interesting and is well founded and developed with a relevant methodology, which is clearly described. It is suggested to enhance the images, for example, in figure 1, which is shown in each image (right and left). Not all images are cited in the document.
It is suggested to add explanatory notes on the figures to improve their understanding. Add an image of the city under study, where it is located in Spain and within it, which is the area studied.
The images have been improved and referenced
It is not clear if all the people who answered the questionnaire live in the historic center, or are distributed throughout the city.
The surveys were online and distributed throughout the city, in order to compare the homes in the historic center with contemporary homes. One of the questions in the survey was the year of construction of the house.
Reviewer 3 Report
This paper adopt the quantitative method in optimizing energy consumption . One suggestion is compare the current data with history data and understanding the difference between lockdown period and non-lockdown period.
Author Response
First, we would like to say thank you to the reviewers for the useful comments to improve the paper. We have addressed all the comments as explained below.
Reviewer 3:
This paper adopt the quantitative method in optimizing energy consumption . One suggestion is compare the current data with history data and understanding the difference between lockdown period and non-lockdown period.
This would be very interesting, but these data are currently unpublished. In the coming months, it will be accessed and will be considered in a future publication.